# AI, ageing and brain-work productivity: Technological change in professional Japanese chess

**Eiji YAMAMURA** [1]*, **Ryohei HAYASHI** [2]

**1** Department of Economics, Seinan Gakuin University, Fukuoka, Japan, **2** Kochi University of Tecnology, Kochi City, Kochi, Japan

* yamaei@seinan-gu.ac.jp

**Data Availability Statement:** The data underlying the results presented in the study are available from Figure (https://figshare.com/search?q=10. 6084%2Fm9.figshare.24935856).

## Abstract

Using Japanese professional chess (Shogi) players' records in the setting where various external factors are controlled in deterministic and finite games, this paper examines how and the extent to which the emergence of technological changes influences the ageing and innate ability of players' winning probability. We gathered games of professional Shogi players from 1968 to 2019, which we divided into three periods: 1968–1989, 1990–2012 (the diffusion of as information and communications technology (ICT)) and 2013–2019 (artificial intelligence (AI)). We found (1) diffusion of AI reduces the impact of innate ability in players performance. Consequently, the performance gap among same-age players has narrowed; (2) in all the periods, players' winning rates declined consistently from 20 years and as they get older; (3) AI accelerated the ageing decline of the probability of winning, which increased the performance gap among different aged players; (4) the effects of AI on the ageing decline and the probability of winning are observed for high innate skill players but not for low innate skill ones. The findings are specific to Shogi as a kind of board games although it is valuable to examine the extent to which the findings hold for other labor market.

## Section 1: Introduction

How will new technological progress, such as information and communications technology (ICT) and artificial intelligence (AI), change the work environment and the labour market? Several studies have tried to answer this question [1–4]. The impact of technological progress varies according to jobs and required skills. For example, the impact of AI on board game players has been remarkable [5, 6]. In 1997, IBM's AI—named Deep Blue—beat Garry Kasparov, the world's chess champion. The rules of the Japanese chess (Shogi) game are more complicated than for traditional chess and the possible choices per game are far higher. Multiple professional Shogi players did not predict that they would be defeated by AI [7]. Owing to the limitations in technology in 2007, AI could not win against Akira Watanabe, who was a top-class professional Shogi player and the major title 'Ryuo' holder. Based on his experience in the game, Watanabe predicted that AI could not win against professional Shogi players. However, the level of AI was almost equivalent to the top-notch amateur Shogi players [8]. After 6 years, in 2013, AI defeated the professional Shogi player, Shin-ichi Sato. Since then until 2017,

**Funding:** The author(s) received no specific funding for this work.

**Competing interests:** The authors have declared that no competing interests exist.

various professional Shogi players have played against AI but could not win in most cases. The results of the AI vs. Human games indicated that AI could surpass professional Shogi players. Akira Watanabe accepted that AI has surpassed the top-rank professional Shogi players and confessed that he became an active user of AI to improve his Shogi skills in games against other human players [9].

Several studies have dealt with how technological progress influences productivity in the labour market [2, 3, 10–13]. In Europe, polarisation trends in the labour market appeared from 1993 to 2010 because of an increase in high and low-paying occupations, whereas middle-paying occupations decreased [14]. Similarly, non-routine tasks have increased in Japan, while routine tasks have decreased from 1960 to 2005. This suggests a long-term labour market polarisation [4]. Technology leads to wage inequality and polarisation in the labour market [1, 12].

The recent development of AI technology seemingly allows it to replicate the human brain. An estimated at 47% of jobs are expected to be at the risk of computerisation [15]. However, this might be over-estimated. Current AI gives us only the illusion. It excels at pattern recognition, but it cannot reason, and does not understand concepts. In Shogi, AI finds good strategies, but so far only human can rationalize why strategies work, and apply them accordingly. AI will replace only "some intellectual work", because it cannot replicate human reasoning yet.

There is an argument that complementarity between labour and robots increases productivity [1]. The skills and technology required for work differ according to jobs, industries, and stages of economic development. Accordingly, the influence of technology diffusion varies according to the settings where workers are confronted. It is valuable to scrutinise how technology diffusion influences workers' performance and the inequality between workers with different skills within the labour market in a specific industry by comparing different periods.

In a professional board game, various factors that influence labour productivity can be controlled under the setting in deterministic and finite games, especially in a two-player game such as chess, where a player's performance does not depend on a team mate's performance. During the game, players cannot receive the advice of others, or use technological devices. Innate ability, a trait or characteristic present at birth, is one of the key factors determining individual performance and, hence, labour productivity. Apart from it, human capital is accumulated through learning from experience, which improves productivity, whereas mental ability declines with age, which lowers productivity. Researchers examined how chess instruction changes children's educational attainment. In developed countries, the effect is not observed [16], whereas, in developing countries, the positive effect is observed to a certain extent [17].

The relationship between age and performance is an inverted U-shape profile with a peak at approximately 21 years for chess players' productivity [19]. Ageing players consider AI more difficult to use and are less likely to catch up with drastic environmental changes. However, it is unknown whether technological progress, such as ICT and AI, influences innate ability and ageing on player's performance.

We constructed a database of games from 1968 to 2019 to compare ageing and innate ability effects between three sub-periods. From the data, the major findings are that performance polarisation is observed in the period when AI is widely used to improve strategies among professional Shogi players. The following mechanism explains this. AI training reduces the importance of the innate ability of an individual player, which reduces the gap in performance between players of the same age. However, the effects of negative ageing on performance are strengthened by the diffusion of AI, inducing players to retire from active play earlier than in other periods. These imply that the polarisation of players' performance depends on whether players can make the best use of AI rather than innate ability.

The remainder of this article is organised as follows. Section 2 overviews the real situation of professional Shogi. Section 3 explains the dataset and presents the basic statistics. Section 4

proposes testable hypotheses and describes the empirical method. Section 5 presents and interprets the estimated results. In Section 6, we provide discussion about relation between AI and labour market. Section 7 offers some reflections and conclusions.

## Section 2: Overview of professional Shogi

### 2.1. Rules and differences from international chess

On some points, the rules of Shogi are the same as that of international chess. The game's goal is for one player to checkmate the other player's king, winning the game. Several Shogi pieces can be moved like those of international chess. However, Shogi is different from international chess as follows. In cases where a piece occupies a legal destination for an opposing piece, it may be captured by replacing it with the opposing piece. Captured pieces are retained in hand and can be brought back into play under the capturing player's control. On any turn, instead of moving a piece on the board, a player may select a piece in hand and place it on any empty square. Therefore, one of that player's active pieces on the board can be moved. This is called dropping the piece or simply, a drop. A drop counts as a complete move. The ability to drop in Shogi drastically increases the player's choices, which leads to more strategic variety and complexity than in international chess. Naturally, AI won against a professional Shogi player 16 years later than Deep Blue won against Kasparov in 1997.

The differences in the rules are the reason why the Shogi game rarely ends in a draw. In the professional Shogi games from 1968 to 2019, the rate of draw games was 0.7% which is far lower than the 53.4% draw rate for international Chess from 1970 to 2017 [18] The index of a player's performance is made by wins, losses, and draws, owing to the extremely high draw rates in previous international chess studies [19]. However, as opposed to chess, Shogi players are unlikely to develop a strategy to draw. In this study, it is not necessary to consider draw rates because draws rarely occur in Shogi.

### 2.2. System

Shogi players are promoted to professional status from the 'Shoreikai League', where young amateur players are selectively qualified to enter. Regardless of gender, promising amateurs can apply for the 'Shoreikai'. Members of the Shoreikai are neither professional players nor amateur players because they all aspire to survive the fierce struggle for existence to become professional players. The Shoreikai consists of approximately 200 players. Players younger than 15 years qualify for an entrance examination held once a year in August. In the first stage, a candidate plays games with 6 other candidates. They proceed to the second stage if they gain a majority of wins. In the second stage, the candidate plays games with 3 incumbent Shoreikai members. The candidate passes the examination if the candidate wins at least one game. There are 30–40 candidates and the ratio of successful applicants is approximately 10%–20% [19]. In most cases, Shoreikai players start their career from the bottom grade 6-kyu directly after passing the entrance examination. There are 9 grades in Shoreikai, and the top grade is 3-dan. The higher the grade is, the grade number of 'kyu' reduces. That is, starting from 6-kyu, 5- kyu, 4-kyu, 3-kyu, 2-kyu, 1-kyu. After 1 kyu, the grade number of 'dan' increases. That is, 1-dan, 2-dan and 3-dan.

Members can proceed to the next higher grade if they gain high winning rates in the league of each grade. Approximately 30 members constitute the top league, the 3-dan league. Among them, 2 members can become professional players in half a year only if they gain the first or the second position in the 3-dan league. After successfully being winners in the final stage of Shoreikai, they become 4-dan grade, which means they can enter the professional league. Therefore, every year, only four Shoreikai members can become professional players. Further,

members must withdraw from the Shoreikai, regardless of their intention, at 26 years and cannot proceed to the professional level. The Shoreikai system forms a straight gate to enter the world of professional Shogi, even though most Shoreikai players cannot survive and drop out from the Shoreikai league before becoming professional players.

The number of professional players is kept constant. Hence, four professional players retire every year to be replaced by four players promoted from the Shoreikai. As a retirement rule, professional players are forced to retire if their winning rate has been extremely low for several years. However, it is not difficult to keep their professional status once they become professional players. There were 301 professional Shogi players from 1968 to 2019. Only 7 players have retired below 45 years, owing to their poor performance. Thus, most of them could play as professionals for 45 years even if their performances were low. The winning rate of professional players declines as they get older, and naturally, they retire in their 50s or 60s. Consequently, below 45, a professional players' perfect record can be obtained from their debuts. This is a setting unique to professional Shogi players, different from the international chess league, where teenagers can participate and drop out frequently [20]. Therefore, members of the professional league hardly change before they turn 45 years old because they survive in the highly competitive Shoreikai league before becoming professional players.

Membership to the Shoreikai is open to male and female players. However, no female player has yet accomplished this feat. Females are not able to win because of their lack of skill. Thus, all professional players are males, partly because of the very competitive environment in the Shoreikai and the gender difference in performance in a mixed-gender competitive environment [21]. No female Shogi players can win through in Shoreikai and be promoted to become professional players. The situation is different from chess where male and female compete in games [22–27]. There is a separate and different system specially designed for female professionals. Therefore, some female Shorei kai players can also play the game as 'Female Professionals' who do not have the right to participate in the professional league (Jun-i Sen League).

After becoming professionals, Shogi players usually enter the league to play for the Meijin title, called 'Jun-i Sen'. In this league, there are 5 classes: 'C2 (bottom)', 'C1', 'B2', 'B1', 'A (top)'. Similar to Shoreikai, only the first and second ranks of winning rates in a class can be promoted to a higher class. The last and second to the last rank players automatically move to the lower class. The change of class occurs once a year and is considered to reflect the player's strength. It takes at least 5 years to become a class 'A' member after entering the professional league. The number of class 'A' players is fixed at 10. The player with the highest winning rate in the season has the right to play with the title holder of Meijin. Therefore, a player cannot play with the title holder of Meijin if he does not belong to 'A' even if he is stronger than any 'A' class player. Apart from 'Meijin', there are 7 other titles, which include 'Ryuo', 'Oi', 'Oza', 'Kio', 'Eio', 'Osho' and 'Kisei'. Unlike Meijin, any professional player could get these titles if they survive the fierce struggle in the title tournament. Among 8 major titles, 'Meijin' and 'Ryuo' are equally the highest status because of the longest history of 'Meijin' and the highest prize money of 'Ryuo'. Besides the league and tournaments for the 8 major titles, there are different games for some minor titles.

Besides their class in the league, the status of players can be captured by 'dan'. In the professional world, there are 6 player grades: 4, 5, 6, 7, 8 and 9-dan. This reflects the experience of the total number of winnings from their debut. Hence, players' grades increase as they gain experience in games. However, the grade decreases if their performance is very low. Meanwhile, the degree of dan never decrease once the dan was certificated to the player. That is, the degree of dan is fixed even if player's winning ratio reduced drastically. Therefore, the degree of dan is unlikely to reflect the players' strength at the present time.

## 2.3. Features of the three periods

Accurate data is available from 1968, therefore, we limited the data before 1968 to avoid measurement errors. The professional Shogi environment from 1968 to 2019 can be divided into three periods in terms of technological development.

First, 1968–1989 was characterised by human-led Shogi before technology emerged. This period is equivalent to the later Showa era in Japan. Shogi players trained themselves to learn from books, including the records of historically famous professional games before ICT emerged. Hence, players' strategies and skills developed slowly. Rather than improving skills and strategies, players' real-life experience is considered important to win in a game because the psychological tactics of the game effectively destroy the opponents' morale. For instance, most players feel mental stress before a game, decreasing their appetite. In contrast, Yasuharu Oyama, a multiple title holder for many years, would intentionally eat a lot in front of his opponent before a game to display his toughness. The opponent often fell victim to Oyama's tactics and could not fully demonstrate his ability.

Second, the period 1990–2012 is characterised by the information-oriented Shogi. ICT had developed during this period covering the early and middle of the Heisei period. Players can search for the latest games records by using a computer database. Further, players frequently met together in the real world and exchanged opinions and investigate new strategies. Naturally, new strategies developed frequently and diffused earlier among professional players than in the Showa era. Players' skills and strategies drastically improved, incapacitating the Oyama tactics style and prevented players from adopting the same tactics. During this period, Yoshiharu Habu, who held multiple titles consistently, argued that 'Shogi is a pure board game and so real life experience does not contribute to a player's performance at all'.

Third, the period 2013–2019 is characterised by the AI-dominant Shogi. As explained in the introduction, in 2013, a professional Shogi player was first defeated by AI. Within the next several years, various top-ranked players also lost games to AI. In 2017, there were two games of AI vs. Amahiko Sato, who held the title of 'Meijin'—the most valuable title. Amahiko Sato suffered complete defeats in both games, implying that AI exceeded professional players. According to the major multiple title holder, Akira Watanabe, 'Before diffusion of AI, it took around 10 years to establish a standard move, whereas it takes only a day now. This results in an unexpected situation. Any players can use AI, leading to the gap in Shogi skills between players to be narrowed' [9]. Owing to AI's diffusion, top-rank players are less likely to derive advantage from their innate ability.

## Section 3: Data

We collected game-level data from the Shogi database through the internet [28]. The data information includes the game's date and results, kinds of game (name of the tournament, class of the league, final and semi-final), player's information, such as name, birth date, age of his debut as a professional player, Elo rating which is the index for player's strength, class to which they belong and grade (dan). The games database covered records from the 1950s. However, in the early period of the database, some games had errors or insufficient information. From 1968, information on games was accumulated. In previous Chess studies, the Elo rating is generally used to measure individual player's strength [24, 29, 30].

Fig 1 demonstrates the Elo rates based on the three different periods defined in section 2. We observe a unimodal distribution for the periods 1968–1989 and 1990–2012. However, from 2013 to 2019, we observe a peak at 1600 and a small bump slightly over 1800. From Fig 1, we infer that in the period of AI, the strength of players is polarised.

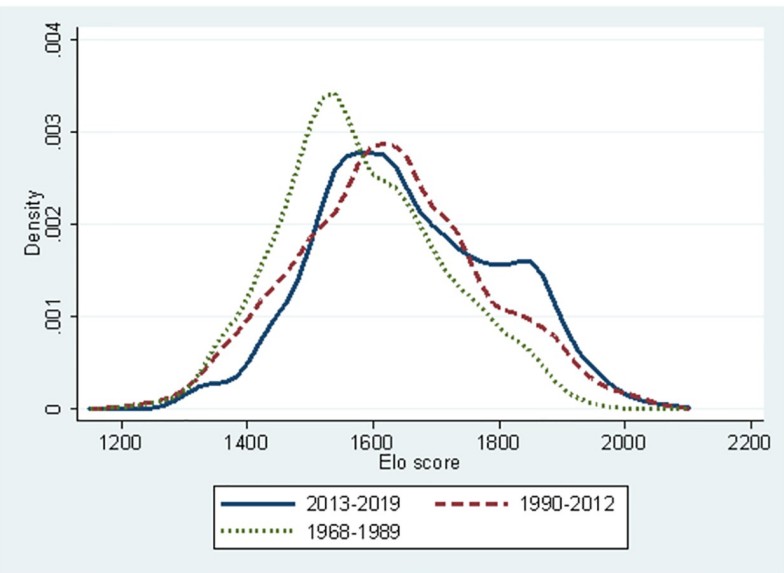

**Fig 1. Kernel distribution of performance score (Elo rate): Period 1968–2019.**

In Fig 2, the sample is divided into the very recent period 2018 to 2019 and the period 1968 to 2017 because AI has been widely diffused and utilised by players since 2017 [9]. Twin peaks are observed from 2018 to 2019, whereas standard distribution is observed from 1968 to 2017. Strength polarisation appeared probably because some players became active users of AI for their investigation of Shogi, whereas others did not or could not effectively use AI. It is crucial whether players could catch up with the drastically changed environment.

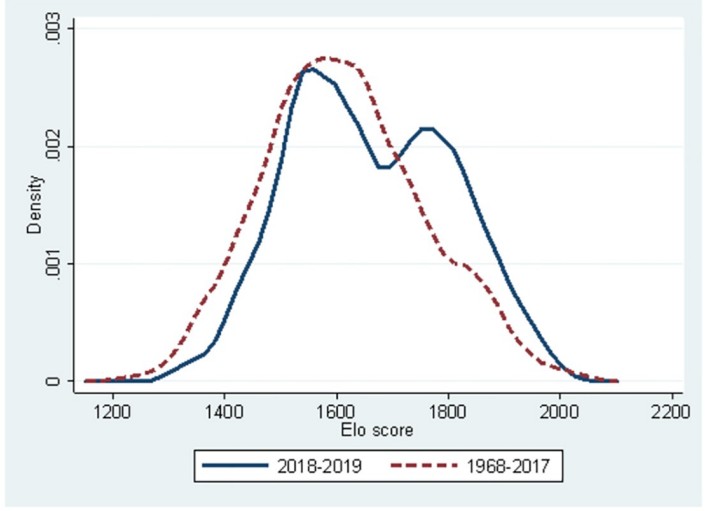

**Fig 2. Kernel distribution of performance score (Elo rate): 1968–2017 vs. 2018–2019.**

**Table 1. Definitions of key variables and their basic statistics.**

| Variables | Definition | Mean | s.d. | Min. | Max. |
|---|---|---|---|---|---|
| *Dependent Variables* | | | | | |
| WIN | Equals 1 if the first players win, 0 otherwise. | 0.53 | 0.49 | 0 | 1 |
| *Independent Variables* | | | | | |
| AGE | The first players' ages | 36.7 | 11.5 | 15 | 76 |
| AGE_OP | The second (opponent) players' ages | 37.8 | 11.8 | 15 | 77 |
| DEB_AGE | The first players' debut ages | 20.9 | 2.65 | 15 | 26 |
| DEB_AGE_OP | The second (opponent) players' debut ages | 21.3 | 3.52 | 15 | 26 |
| ELO/100 | The first players' Elo Score/100 | 16.2 | 1.41 | 11.6 | 20.9 |
| ELO_OP/100 | The second (opponent) players' Elo Score | 16.2 | 1.45 | 11.7 | 20. |

Note: Draw games are excluded from the sample. Sample covers players of all ages, and its observations are 88,788

The Elo rating is considered to be different from performance. 'ELO cannot be considered as a measure of productivity at Chess, which depends on realised rather than unexpected wins and draws. . . Rather than a measure of productivity, ELO is a measure of relative ability at Chess at a given point in time: it predicts ex-ante how likely a player is to win when he plays against an opponent, but it does not measure winning intensity' [20]. However, the Elo rating changes to reflect the results of games after becoming professional players. Therefore, it captures the innate ability and learning effect from the player's experience.

As explained in section 2, it is difficult to enter a professional Shogi league, unlike in international Chess. During the 1968–2019 period, all the titleholders of the Meijin debuted as professional players when they were teenagers, younger than the mean debut age of approximately 21 years, as shown in Table 1. Hence, the younger the player started, the higher their innate ability. Under the setting of the professional Shogi system, the innate ability can be captured by the debut ages of the professional players. Further, similar to international Chess, self-selection bias possibly occurs in professional Shogi [20]. However, only 0.7% retired below 45 years in the studied period owing to their poor performance. Suppose we restrict the sample to professional players; the self-selection bias is unlikely to influence the results. This study uses a full sample covering aged players and a sub-sample limited to players younger than 45 years.

Fig 3 illustrates how total winning rates change according to players' debut age. Using the full sample, we observe that the average debut age winning rate is approximately 0.7, implying that the rookie professional player's winning rate is approximately 70%. Players' winning rate reduced to approximately 50% when they debuted at 22 years and further declined slightly below 40% when they debuted at 26 years. Winning rates tend to decline as debut age increases, using the full and the sub-samples below 45 years. There is no statistically significant difference between the samples. The opponent's strength changes according to age because players with higher performance are likely to play with similar level players to compete for a major title in a higher class league.

Fig 4 demonstrates that the winning rate consistently declines as players become older. The winning rate based on the sub-sample below 45 is lower than that of the full sample. In the sub-sample, all players are below 45 years; therefore, games between players over 45 and below 45 are not included. Game records where younger and older players played with lower winning rates are excluded. Further, the self-selection effect hardly occurs when the sub-sample is used, reducing upward bias. In a previous study, the opponent's strength is controlled using

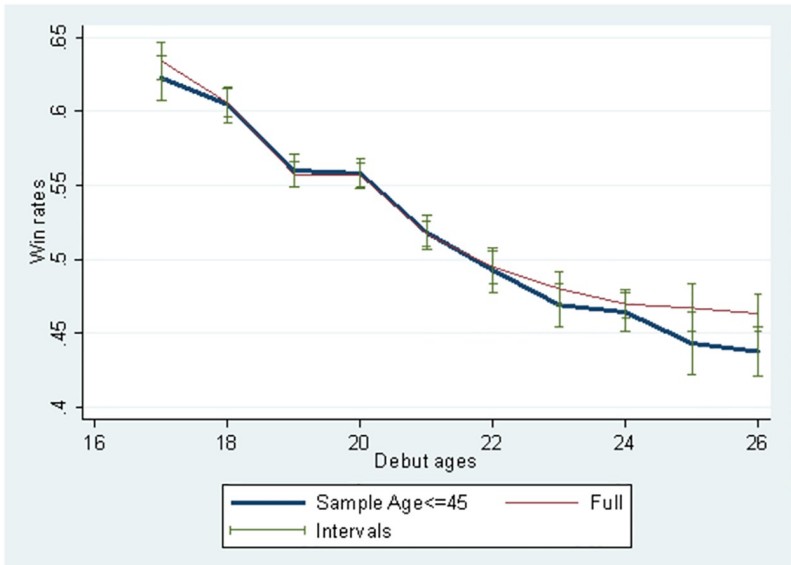

**Fig 3. Change in player's mean win rates according to debut ages.**

Elo ratings [20]. In S1 Fig in the S1 File we can observe a similar we observed a similar trend after controlling the opponent's Elo rating, consistent with previous work [20].

Figs 5 and 6 use the full sample to compare the three periods. Fig 5 illustrates debut ages and winning rates relation. A decline in the total winning rate was observed in 1968–1989 and 1990–2012. However, in the period 2013–2019, the gap in total winning rate between debut ages is smaller than in the other periods, although the winning rate slightly declines as debut age gets older. Therefore, the impact of the innate ability on winning rate is smaller in the

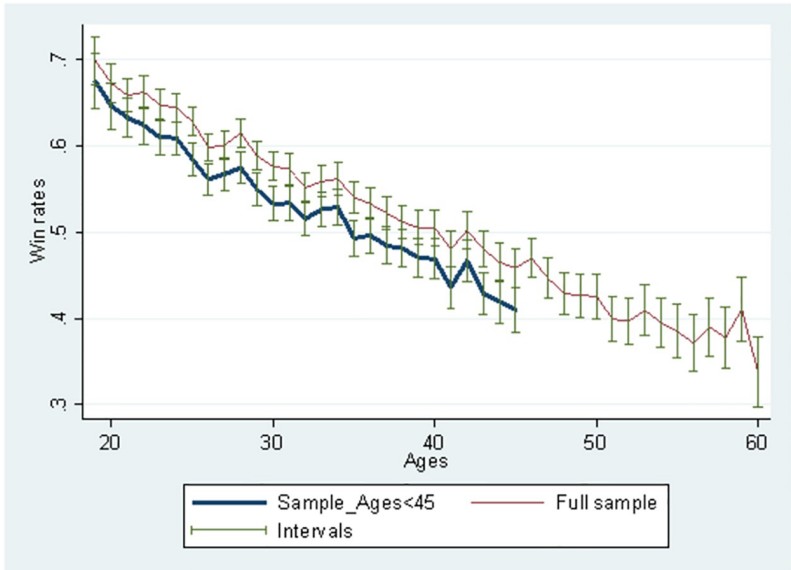

**Fig 4. Change in player's mean win rates according to ages.**

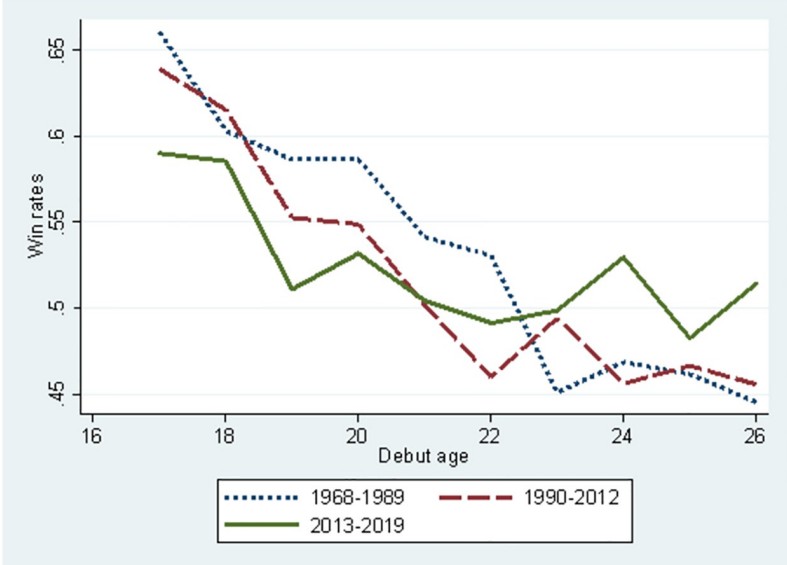

**Fig 5. Comparison between the three periods about change in player's mean win rates according to debut ages.**

period when skill and strategy improved drastically. The diffusion of AI reduces the advantage of innate ability because players with lower innate abilities utilise AI to catch up with those with higher innate abilities. AI can be used to reduce the gap in innate abilities between players. Fig 6 illustrates the ages and the winning rates relationship. The winning rate constantly declined during the 1968–1989 period as players got older. However, winning rates hardly changed for players older than 50 years. There are two possible reasons why the winning rates remained constant at an advanced age. First, players learn the psychological tactics of the game

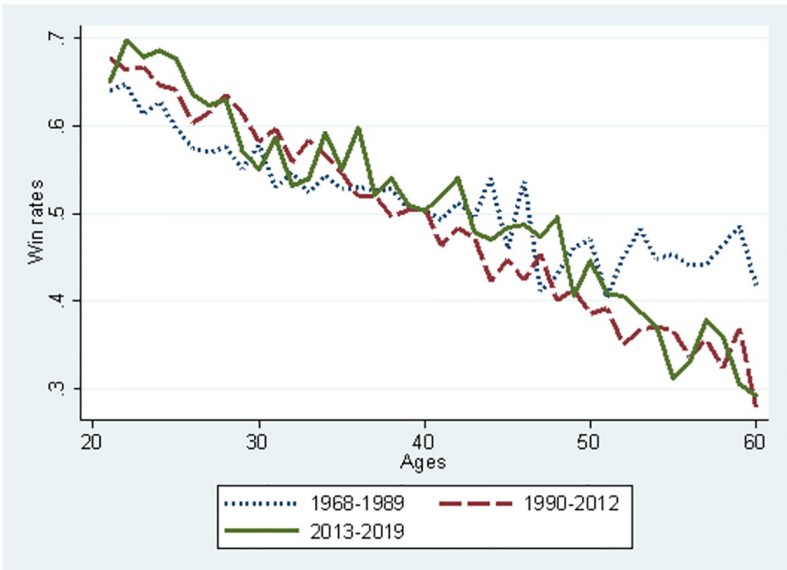

**Fig 6. Comparison between the three periods about change in player's mean win rates according to ages.**

through their experiences, which effectively maintains their winning rates. Second, more able players can survive despite their advanced ages, whereas other aged players are forced to retire owing to poor performance. During the 1990–2012 and 2013–2019 periods, the decline in the total winning rate is constantly observed. Even in these periods, the self-selection effect existed but did not maintain the aged player's winning rate. Overall, the three periods jointly suggest that drastic improvement in skills and strategies owing to ICT or AI reduces the effect of learning psychological tactics, resulting in a consistent decline in the winning rates.

## Section 4: Hypothesis and method

### 4.1. Hypotheses

In Figs 1–6, we can observe that players' performance changes. From these observations and the environmental change of professional Shogi, we propose several testable hypotheses in this section. Following Fig 1 and the findings from international Chess [20], mental productivity of board games declines approximately after 20 years; we propose *Hypothesis 1*.

*Hypothesis 1. Winning rates decline as players get older after their debut.*

The diffusion of AI distinctly accelerates skills and strategy development. Therefore, older players face a challenge catching up with environmental change after the diffusion of AI. We propose *Hypothesis 2*.

*Hypothesis 2. Declining of winning rates due to ageing is more rapid after AI diffusion than ever before.*

### 4.2. Method

In the model, the estimated function takes the following form to examine how ages and innate ability influence players' performance. We use the winning dummy for a game (WIN) as a proxy variable for an individual's performance and dependent variable in the model. The Probit model is used for this estimation because the dependent variable is 0 or 1.

$$
\begin{aligned}
\text{Prob}\Big[\text{WIN}_{igt} = 1\Big] \quad &= F(\alpha_0 + \alpha_1\,\text{AGE}_{igt} + \alpha_2\,\text{AGE}^2_{igt} + \alpha_3\,\text{AGE\_OP}_{igt} + \alpha_4\,\text{AGE\_OP}^2_{igt} \\
&\quad + \alpha_5\,\text{DEB\_AGE}_i + \alpha_6\,\text{DED\_AGE}^2_i + \alpha_7\,\text{DEB\_AGE\_OP}_i \\
&\quad + \alpha_8\,\text{DED\_AGE\_OP}_i^{\,2} + \alpha_9\,\text{ELO}_{igt} + \alpha_9\,\text{ELO\_OP}_{igt} + X'_{it}B).
\end{aligned}
$$

where F denotes the cumulative distribution function of the standard normal distribution. In the function, the suffix '*i*' is an individual player, suffix '*g*' is a game, and suffix '*t*' is the time point. WIN is the first player's winning dummy which is 1 if the first player wins; otherwise, 0. Table 1 indicates that the mean value of WIN (win dummy) is 0.53, implying that the first player's winning rate is 53%, while the second player's is 47%. Therefore, the first players enjoy some advantage, although the difference between the first and second players is small. In professional Shogi, it is randomly determined whether a player is first or second. Hence, as shown in Table 1, the first player's age in the game and his debut age are similar to those of the second player.

As explained in section 2, unlike international Chess, the draw rate is only 0.7%, so draw games are not included in the sample. Compared to international Chess, a player's productivity can be captured by a player's win because it is unnecessary to consider a draw. However, it is critical to control the opponent's and player's Elo ratings. Furthermore, the player's

productivity is calculated by considering the opponent's Elo rating [20]. As seen in the functional from of our model, this study includes both Elo ratings as independent variables. Further, it is more convenient to interpret the results when the probability of winning is investigated.

This study examines the relationship between players' ages and their performances. Shogi is a two-player game; we should consider both players' characteristics. The opponent (second player) with lower ability increases the wining probability of the first player. Therefore, the predicted sign of the first player's characteristics is opposed to that of the second player. The ages of the first player (AGE) and that of the second player (AGE_OP) are included as key variables. From *Hypothesis 1*, the expected signs of AGE and AGE_OP are negative and positive, respectively. To capture first the players' innate ability and that of his opponent, the debut age of the first (DEB_AGE) and second player (DEB_AGE_OP) are included. The older the debut age, the lower the innate ability. Hence, the expected coefficient's sign is negative and positive for DEB_AGE and DEB_AGE_OP, respectively. The relationship between the WIN and these variables is potentially non-linear. For this reason, we incorporated their squared terms such as $DEB\_AGE^2$, $DEB\_AGE\_OP^2$, $AGE^2$, and $AGE\_OP^2$. However, for convenience of interpretation, their results of marginal effect are divided by 100. Therefore, ELO and ELO_OP are incorporated, and their expected sign is positive and negative, respectively.

$X_i$ represents the control variable vector, and B is the vector of their coefficients. X consists of the time point of the game, dummies for the first and the second players' class of the league, dummies for the first and the second players' 'dan'. These variables also control for players' strength. Besides it, we add dummies for the status and classification of games because the importance of games widely varies. For instance, it is much more important to win a championship game of a major title than a preliminary game. In addition to the coefficient, we also report the marginal effect of the independent variables at their means for convenience of interpretation. The marginal effect in the probit model is expressed below. We assume that the specification with the quadratic term is

$$Pr(Y = 1|X) = F(\alpha_0 + \alpha_1 X + \alpha_2 X^2).$$

The coefficient of X is $\alpha_1$, while that of $X^2$ is $\alpha_2$. In addition to these coefficients, a marginal effect can be obtained at *X*,

$$\frac{\partial Pr(Y = 1|X)}{\partial x} = (\alpha_1 + \alpha_2 2X) \times F'(\alpha_0 + \alpha_1 X + \alpha_2 X^2)$$

The value of *X* varies from 20 to 60, if *X* is "Ages" using a sample covering 20–60 years. Therefore, apart from the mean value of ages, we can estimate 40 marginal effects of "Age." Furthermore, for the independent variable, even without the quadratic term, we obtain the marginal effect, as follows:

$$Pr(Y = 1|X) = F(\alpha_0 + \alpha_1 X).$$

The marginal effect can be obtained at *X*,

$$\frac{\partial Pr(Y = 1|X)}{\partial x} = \alpha_1 \times F'(\alpha_0 + \alpha_1 X)$$

We obtain a value of $\alpha_1$ being a coefficient of X. However, we can calculate various marginal effects according to value X. In the S1 Figs 2–7 in the S1 File illustrate marginal effects with 95% confidence intervals for each specification.

Furthermore, based on the marginal effect of the probit model results in each period, we provide an analysis of predictions to visually present how a player's performance differs with respect to his innate ability and how a player's performance changes as he gets older. Player performance is the winning rate. We then compare visualised changes in performance between three periods to consider how the path of a player's performance declination differs according to technologies such as ICT and AI.

## Section 5: Results

### 5.1. Results of the Probit model

Tables 2–5 show the Probit model results. In Panels A and B, the values without parentheses for each variable are marginal effects but not coefficient and marginal effects at their means, respectively. In addition, marginal effects of AGE at 35 years is reported on the bottom line. For estimations, we used Stata 14 that is widely used in the field of economics. In the program, to take, for example, "AGE," coefficient and marginal effects are calculated as follows:

probit WIN c.AGE##c.AGE

margins, dydx(*) atmeans

margins, dydx(AGE) at(AGE = 35).

We provided the program as a S1 File.

Table 2 shows the results based on the sample covering all ages. The results of Table 3 are based on the sub-sample restricting players who are 45 years or below to mitigate the self-selection biases. Further, the sub-sample is divided into players who debuted at 20 years or below and players who debuted above 20 years. Table 4 presents results using the sub-sample of players who debuted at an early age, considered to have high innate ability. Table 5 indicates results using the sub-sample of players who debuted at a later age, considered to have low innate ability.

Firstly, from Table 2, we found that the results of most variables show the expected sign. Furthermore, in most variables, statistical significance is observed in columns (1) and (2), whereas statistical significance is not observed in many of the variables in column (3). Looking closely at Column (3) of Panel A shows that AGE and AGE_OP are statistically significant, but their squares are not. This implies that the relationship between ages and the winning rate is decreasing regardless of the other variables in the 2013–2019 period, while the relationship is negative and non-linear before the diffusion of AI. In the 2013–2019 period on Panel B, The marginal effect of AGE is −0.022 or −0.023, implying that the first player's probability of winning is reduced by approximately 2.2% if he gets a year older. However, such reduction in winning rate is mitigated as players become older in the periods before AI diffusion because the marginal effects are −0.004 and −0.006 for the 1968–1989 and 1990–2012 periods, respectively.

Regarding DEB_AGE and its square (DEB_AGE$^2$), these show significant expected signs in columns (1) and (2) but not in column (3). Meanwhile, DEB_AGE_OP and its square (DEB_AGE_OP$^2$) indicate the predicted significant sign in columns (3) but not in columns (1) and (2). Consistent with this prediction, ELO and ELO_OP show the expected positive and negative signs, respectively, which are statistically significant at the 1% level in all columns.

In Panel A of Table 3, the variable of player's age shows the predicted sign and statistical significance, except for AGE,$^2$ in column (3). The variable of debut ages show the expected signs, although not statistically significant for some of them in columns (1) and (2). The results for the variable of Elo rating is similar to that in Table 2. Overall, the results in Panels A and B of Table 3 are almost equivalent to those of Table 2.

**Table 2. Dependent variable: WIN (Probit model): Sample covered all ages.**

Panel A. Coefficient

| | (1)<br>*1968–1989* | (2)<br>*1990–2012* | (3)<br>*2013–2019* |
|---|---|---|---|
| AGE | − 0.019***<br>(0.005) | − 0.025***<br>(0.004) | − 0.028***<br>(0.008) |
| AGE$^2$/100 | 0.014**<br>(0.005) | 0.015***<br>(0.004) | 0.015<br>(0.009) |
| AGE_OP | 0.022***<br>(0.004) | 0.035***<br>(0.004) | 0.023***<br>(0.008) |
| AGE OP$^2$/100 | − 0.018***<br>(0.004) | − 0.029***<br>(0.004) | − 0.014<br>(0.009) |
| DEB_AGE | −0.073***<br>(0.015) | −0.058***<br>(0.018) | −0.042<br>(0.032) |
| DEB_AGE$^2$/100 | 0.134***<br>(0.027) | 0.114***<br>(0.041) | 0.098<br>(0.070) |
| DEB_AGE_OP | 0.008<br>(0.016) | 0.026<br>(0.021) | 0.044*<br>(0.025) |
| DEB_AGE_OP$^2$/100 | 0.001<br>(0.003) | −0.046<br>(0.047) | −0.080*<br>(0.005) |
| ELO/100 | 0.244***<br>(0.012) | 0.256***<br>(0.009) | 0.228***<br>(0.017) |
| ELO_OP/100 | −0.252***<br>(0.011) | −0.238***<br>(0.009) | −0.248***<br>(0.018) |
| Pseudo R$^2$ | 0.08 | 0.10 | 0.10 |
| Observations | 33,212 | 43,077 | 12,499 |

Panel B. Marginal Effects (evaluated at mean values for the independent variables)

| | (1)<br>*1968–1989* | (2)<br>*1990–2012* | (3)<br>*2013–2019* |
|---|---|---|---|
| AGE | − 0.004***<br>(0.0004) | − 0.006***<br>(0.0004) | − 0.007***<br>(0.0007) |
| AGE_OP | 0.003***<br>(0.0004) | 0.006***<br>(0.0004) | 0.005***<br>(0.0006) |
| DEB_AGE | −0.005***<br>(0.014) | −0.004***<br>(0.001) | −0.0005<br>(0.002) |
| DEB_AGE_OP | 0.003***<br>(0.001) | 0.003**<br>(0.001) | 0.003<br>(0.002) |
| ELO/100 | 0.095***<br>(0.005) | 0.101***<br>(0.004) | 0.091***<br>(0.007) |
| ELO_OP/100 | −0.100***<br>(0.004) | −0.094***<br>(0.004) | −0.098***<br>(0.007) |
| AGE (Age at 35) | − 0.004***<br>(0.0004) | − 0.005***<br>(0.0004) | − 0.006***<br>(0.0006) |

Note:

\*\*\*, \*\* and * denote statistical significance at the 1%, 5%, and 10% levels, respectively. Numbers without parentheses are marginal effects. Various control variables are included in all columns, such as dummies for a player's and his opponent's rank from 4 dan to 9 dan, the game's status dummies, and the year when the game was held. In addition, game status dummies are made based on the following tournaments or leagues: 8 major title-match leagues (or tournament) (1) Ryuo, (2) Meijin, (3) Oi, (4) Oza, (5) Kio, (6) Eio, (7) Osho, (8) Kisei. Non-major title tournaments such (9) Asahi-hai, (10) NHK-hai, (11) Ginga-sen, (12) Japan Professional-hai (13) Shinjin-o, (13) YAMADA challenge-hai, (14) Kakogawa-sen. However, these estimates are not reported.

In Panel B, the marginal effects are evaluated at the mean values for the independent variables, except for the bottom line, where the marginal effect of age is evaluated at age 35.

**Table 3. Dependent variable: WIN (Probit model): Sub-sample of players below 45 years old.**

Panel A. Coefficient

|  | (1) 1968–1989 | (2) 1990–2012 | (3) 2013–2019 |
|---|---|---|---|
| AGE | − 0.059*** (0.015) | − 0.055*** (0.011) | − 0.055** (0.024) |
| AGE$^2$/100 | 0.077*** (0.022) | 0.059*** (0.017) | 0.056 (0.036) |
| AGE_OP | 0.064*** (0.015) | 0.047*** (0.012) | 0.107*** (0.022) |
| AGE OP$^2$/100 | − 0.075*** (0.023) | − 0.044** (0.019) | − 0.143*** (0.032) |
| DEB_AGE | −0.064* (0.037) | −0.061** (0.026) | −0.033 (0.027) |
| DEB_AGE$^2$/100 | 0.113 (0.081) | 0.127** (0.059) | 0.088* (0.049) |
| DEB_AGE_OP | −0.018 (0.032) | 0.053* (0.027) | 0.075* (0.044) |
| DEB_AGE_OP$^2$/100 | 0.045 (0.072) | −0.109* (0.059) | −0.145 (0.095) |
| ELO/100 | 0.243*** (0.018) | 0.261*** (0.010) | 0.250*** (0.023) |
| ELO_OP/100 | −0.235*** (0.015) | −0.232*** (0.011) | −0.212*** (0.025) |
| Pseudo R$^2$ | 0.07 | 0.08 | 0.07 |
| Observations | 17,881 | 27,933 | 6,416 |

Panel B. Marginal Effects

|  | (1) 1968–1989 | (2) 1990–2012 | (3) 2013–2019 |
|---|---|---|---|
| AGE | − 0.004*** (0.001) | − 0.006*** (0.0006) | − 0.007*** (0.001) |
| AGE_OP | 0.006*** (0.0008) | 0.007*** (0.0007) | 0.006*** (0.001) |
| DEB_AGE | −0.006*** (0.002) | −0.003** (0.001) | 0.001 (0.003) |
| DEB_AGE_OP | 0.0003 (0.002) | 0.003* (0.002) | 0.0006* (0.004) |
| ELO/100 | 0.096*** (0.007) | 0.103*** (0.004) | 0.098*** (0.009) |
| ELO_OP/100 | −0.094*** (0.006) | −0.092*** (0.004) | −0.083*** (0.009) |
| AGE (Age at 35) | − 0.002** (0.009) | − 0.005*** (0.0006) | − 0.006*** (0.001) |

Note:

***, ** and * denote statistical significance at the 1%, 5%, and 10% levels, respectively. Numbers without parentheses are marginal effects. In all columns, control variables included in estimations of Table 2 are included, although these estimates are not reported.

In Panel B, the marginal effects are evaluated at the mean values for the independent variables, except for the bottom line, where the marginal effect of age is evaluated at age 35.

Even after dividing the sample into high- and low-ability players, as shown in Tables 4 and 5, the signs and statistical significance of the variables were similar to those in Table 3. Regarding the marginal value in Panel B, we found differences between Tables 4 and 5. The absolute marginal effects of AGE in Columns (1)–(3) in Table 4 were larger than those in the corresponding Columns in Table 5. The declining performance of high innate-ability players

**Table 4. Dependent variable: WIN (Probit model): Sub-sample of players below 45.** Sample of High innate ability (DEB_AGE< = 20).

| Panel A. Coefficient | | | |
|---|---|---|---|
| | **(1)** *1968–1989* | **(2)** *1990–2012* | **(3)** *2013–2019* |
| AGE | − 0.072** (0.028) | − 0.040** (0.017) | − 0.086** (0.037) |
| AGE$^2$/100 | 0.098** (0.045) | 0.035 (0.025) | 0.099* (0.056) |
| AGE_OP | 0.063** (0.029) | 0.053** (0.024) | 0.078** (0.030) |
| AGE OP$^2$/100 | − 0.085* (0.043) | − 0.060 (0.038) | − 0.100** (0.043) |
| DEB_AGE | −0.067 (0.284) | −0.114 (0.203) | −0.825* (0.447) |
| DEB_AGE$^2$/100 | 0.185 (0.863) | 0.240 (0.615) | 2.380* (1.281) |
| DEB_AGE_OP | −0.132*** (0.048) | 0.105*** (0.038) | 0.126 (0.087) |
| DEB_AGE_OP$^2$/100 | 0.335*** (0.115) | −0.229*** (0.085) | −0.254 (0.196) |
| ELO/100 | 0.241*** (0.039) | 0.241*** (0.016) | 0.203*** (0.036) |
| ELO_OP/100 | −0.215*** (0.033) | −0.229*** (0.016) | −0.214*** (0.040) |
| Pseudo R$^2$ | 0.06 | 0.07 | 0.06 |
| Observations | 4,686 | 10,995 | 2,586 |
| Panel B. Marginal Effects | | | |
| | **(1)** *1968–1989* | **(2)** *1990–2012* | **(3)** *2013–2019* |
| AGE | − 0.007*** (0.002) | − 0.007*** (0.001) | − 0.010*** (0.002) |
| AGE_OP | 0.004** (0.002) | 0.006*** (0.001) | 0.005** (0.002) |
| DEB_AGE | −0.0004 (0.011) | −0.010 (0.007) | 0.008 (0.008) |
| DEB_AGE_OP | 0.003 (0.004) | 0.005** (0.002) | 0.010* (0.006) |
| ELO/100 | 0.092*** (0.001) | 0.092*** (0.006) | 0.078*** (0.014) |
| ELO_OP/100 | −0.082*** (0.001) | −0.087*** (0.006) | −0.082*** (0.015) |
| AGE (Age at 35) | − 0.001 (0.002) | − 0.005*** (0.001) | − 0.006*** (0.002) |

Note:

***, ** and * denote statistical significance at the 1%, 5%, and 10% levels, respectively. Numbers without parentheses are marginal effects. In all columns, control variables included in estimations of Table 2 are included, although these estimates are not reported.

In Panel B, the marginal effects are evaluated at the mean values of the independent variables, except for the bottom line, where the marginal effect of age is evaluated at the age of 35.

due to aging was more rapid than that of the low innate-ability players. Especially in the 2013–2019 period, the players' probability of winning was reduced by 1% if they aged a year. The negative effect of aging on a player's winning percentage became sizable after the diffusion of AI.

**Table 5. Dependent variable: WIN (Probit model): Sub-sample of players below 45.** Sample of Low innate ability (DEB_AGE>20).

Panel A. Coefficient

| | (1) 1968–1989 | (2) 1990–2012 | (3) 2013–2019 |
|---|---|---|---|
| AGE | − 0.047** (0.023) | − 0.071*** (0.017) | − 0.044 (0.040) |
| $AGE^2/100$ | 0.058* (0.034) | 0.083*** (0.025) | 0.037 (0.061) |
| AGE_OP | 0.060*** (0.018) | 0.045*** (0.014) | 0.121*** (0.031) |
| $AGE\ OP^2/100$ | − 0.066** (0.027) | − 0.037* (0.020) | − 0.163*** (0.046) |
| DEB_AGE | −0.377*** (0.097) | −0.023 (0.047) | −0.075 (0.048) |
| $DEB\_AGE^2/100$ | 0.743*** (0.193) | 0.053 (0.097) | 0.163* (0.084) |
| DEB_AGE_OP | 0.025 (0.040) | 0.015 (0.038) | 0.058 (0.053) |
| $DEB\_AGE\_OP^2/100$ | 0.059 (0.087) | −0.027 (0.083) | −0.113 (0.108) |
| ELO/100 | 0.228*** (0.022) | 0.272*** (0.013) | 0.286*** (0.032) |
| ELO_OP/100 | −0.237*** (0.016) | −0.234*** (0.016) | −0.215*** (0.031) |
| Pseudo $R^2$ | 0.07 | 0.08 | 0.07 |
| Observations | 13,195 | 16,398 | 3,830 |

Panel B. Marginal Effects

| | (1) 1968–1989 | (2) 1990–2012 | (3) 2013–2019 |
|---|---|---|---|
| AGE | − 0.004*** (0.001) | − 0.006*** (0.001) | − 0.008*** (0.001) |
| AGE_OP | 0.007*** (0.001) | 0.008*** (0.001) | 0.008*** (0.002) |
| DEB_AGE | −0.014*** (0.004) | −0.0002 (0.002) | −0.003 (0.004) |
| DEB_AGE_OP | −0.0003 (0.019) | 0.001 (0.002) | 0.004 (0.004) |
| ELO/100 | 0.091*** (0.009) | 0.108*** (0.005) | 0.114*** (0.012) |
| ELO_OP/100 | −0.094*** (0.0006) | −0.093*** (0.006) | −0.086*** (0.012) |
| AGE (Age at 35) | − 0.002** (0.001) | − 0.005*** (0.007) | − 0.007*** (0.002) |

Note:

***, **, and * denote statistical significance at the 1%, 5%, and 10% levels, respectively. The numbers without parentheses are marginal effects. The control variables in the estimations in Table 2 are included in all the columns, although these estimates are not reported.

In Panel B, the marginal effects are evaluated at the mean values for the independent variables, except for the bottom line, where the marginal effect of age was evaluated at age 35.

## 5.2. Predictions from the Probit model

As for the predictions from the Probit model about the relationship between the winning rate and debut age, the path is illustrated from 16 to 26 years. From the results in Table 2, we calculate the predicted values of winning rates and visualise the effects of the debut age on the

winning rates in Fig 7. For a base value, we used the mean of the winning rates of those who debuted at the median debut ages for each period.

The median debut age was 21 in 1968–1989 and 22 in 1990–2012 and 2013–2019. The winning rates differed according to the periods: 0.53 (1968–1989), 0.50 (1990–2012) and 0.63 (2013–2019). Fig 7 shows these winning rates and marginal effects at each debut age, as illustrated in the S1 Fig 2 in the S1 File, and we found that most of the marginal effects were lower than 0. Thus, they were negative. Therefore, debut age's effect on winning rate was negative.

Accordingly, the winning rate showed a downward slope if the marginal effect was statistically significant. All negative marginal effects differed significantly from zero in S1 Fig 2(1) in the S1 File. Therefore, in Fig 7, the line of the winning rate is illustrated consistently as a downward slope in the 1968–1990 period. As shown in S1 Fig 2(2) in the S1 File, the negative values significantly differed from 0 for the 16–22 debut ages, but not for the 23–26 debut ages. Hence, in Fig 7, the line of the winning rate is illustrated as a downward slope for the 16–22 debut ages, despite flattening at the 23–26 debut ages in the 1990–2012 period. Interestingly, the negative values for the marginal effects in S1 Fig 2(3) in the S1 File do not significantly differ from zero at any debut age. Debut age did not influence winning rates during the 2013–2019 period. Hence, in Fig 7, the line of the winning rate is flat for the 2013–2019 period. Similarly, Fig 8 shows subsamples equal to or less than 45 years old. In Figs 9–12, instead of debut ages, the marginal effect was calculated at each age, followed by the line of change for the winning rate as the players aged.

The observation for Fig 8 was almost the same as that for Fig 7. The relationship between debut age and win rates varies according to the period. The slope in the period of AI diffusion (2013–2019) was flat; therefore, no innate ability influenced the players' performance as a consequence of AI diffusion. This is consistent with the results shown in Fig 5. Interestingly, the winning rate at a debut age of 16 years was remarkably higher, and the slope was steeper in the Showa low-technology period (1968–1989) than in the other periods. Therefore, the effect of innate ability is remarkably high due to the lack of technology to reduce the innate ability gap.

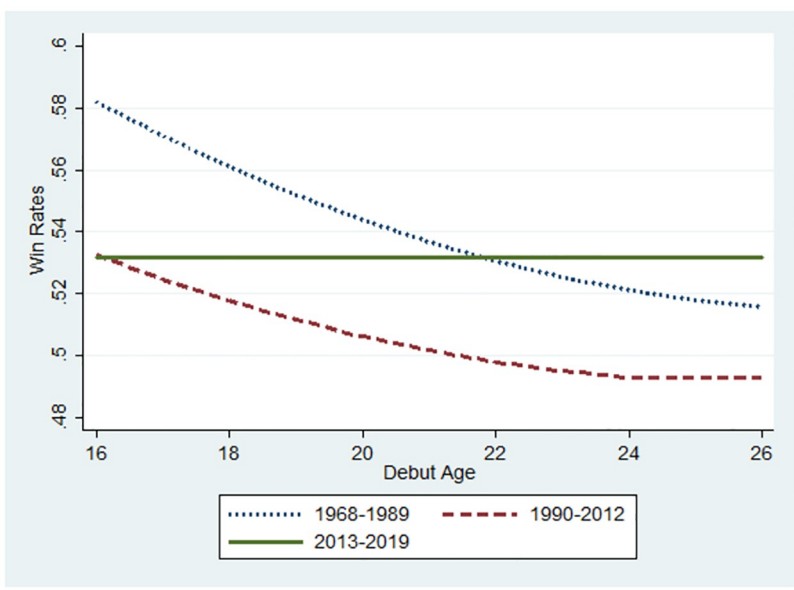

**Fig 7. Predicted values of winning rates and visualise debut age effect on winning rates.**

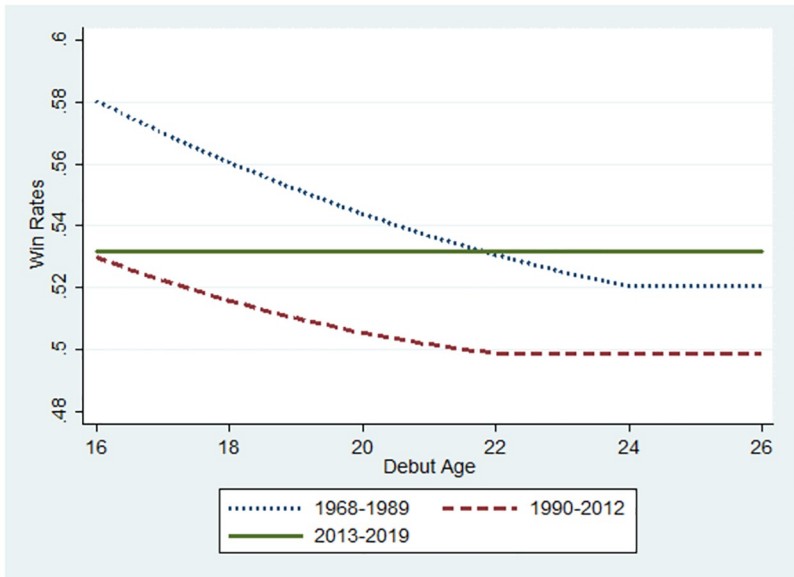

**Fig 8. Changes in performance according to debut ages.** Sample: Ages<45.

The winning rate in the low-technology period was higher than that in the high-technology period (1990–2012) if we compare the winning rates for the same debut ages. The slope for the low-technology period was slightly steeper than that for the high-technology period. Hence, the gap in the winning rate between the two periods decreased when the debut age was higher. Compared to the AI diffusion period, the winning rate gap was remarkably larger in the period before the emergence of AI. This implies that AI contributes to reducing the performance gap between players with high and low innate ability.

Let us now consider the relationship between the winning rate and the players' ages. The base value for the winning rate was the median value for each subsample at 34, 35, and 37 for the 1968–1989, 1990–2012, and 2013–2019 periods, respectively. The marginal effects of age that were statistically significant were used for predictions from the probit model (S1 Figs 4–7 in the S1 File). The initial values of the winning rates and the marginal effects differ according to the three periods. Thus, a player's performance in various settings can be predicted for illustration. Similarly, Figs 7–12 were illustrated by calculating predicted winning rates in each age and period. Figs 9–12 reflect the ageing effect on performance. Fig 9, using the full sample that covers all ages, shows a downward sloping in all the periods. This shows that a player's performance deteriorates over time with a decline in mental strength. Fig 7 demonstrates that winning rates declined as the debut age increased in the three periods. Hence, *Hypothesis 1* is supported, regardless of the environment of the professional Shogi.

The slope in the AI diffusion period is steeper than in any other period. Fig 10 using the sub-sample demonstrates the slope at 45 years because the sample is limited to players who are equal to or below 45 years. In Figs 9 and 10, the slope is steeper for the AI diffusion period than that of other periods, although the winning rate at the debut age of 20 years during the AI diffusion period was the same as that during the ICT period. Naturally, in the AI diffusion period, young players were more likely to win than in other periods, whereas older players were less likely to win. In contrast, in the low technology period, the slope flattened, and its level at the debut age of 20 was lower than that of the two other periods. Therefore,

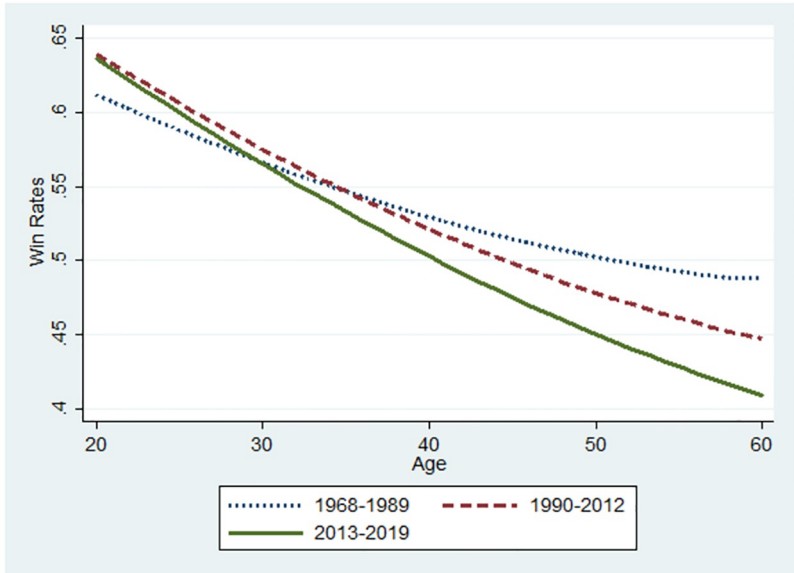

**Fig 9. Changes in performance using the full sample.**

performance gap is smaller among debut ages in the low technology period than that in the later periods. than in other periods. This is consistent with *Hypothesis 2*.

Figs 11 and 12 show the results presented in Tables 4 and 5, respectively. In Fig 11, using a sample of players with high innate abilities, the winning rate at a debut age of approximately 20 years during the AI diffusion period was approximately 0.75, which was the highest among the three periods. However, the gap drastically decreased, and the winning rate during the AI

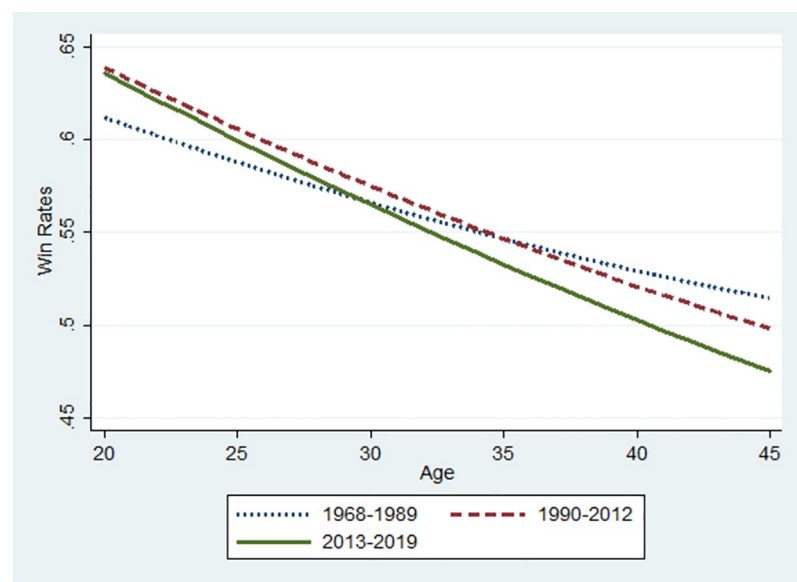

**Fig 10. Changes in performance using the sample below ages 45.**

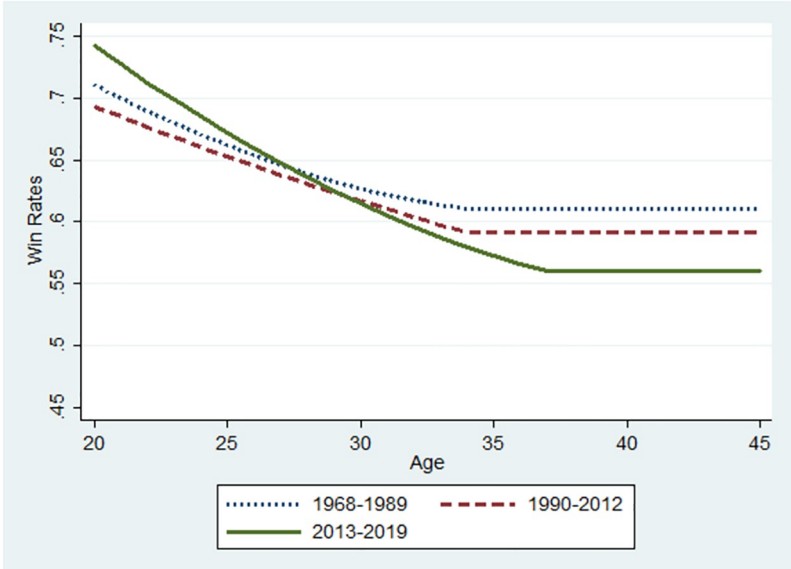

**Fig 11. Changes in performance using the sample of players with high ability.**

diffusion period became the lowest at a debut age of around 30 years because the slope was far steeper than that of the other periods. In Fig 12, all lines are lower than those in Fig 11, thus reflecting the players' low innate abilities. In contrast to Fig 11, the slope during the AI diffusion period was similar to that during the ICT diffusion period. However, the slope in the ICT diffusion period flattened at approximately 37 years, and its winning rate became the same as that in the low-technology period after 37 years. However, the slope in the AI diffusion period became flatter at 40 years, and the winning rate over 35 years was the lowest among the three

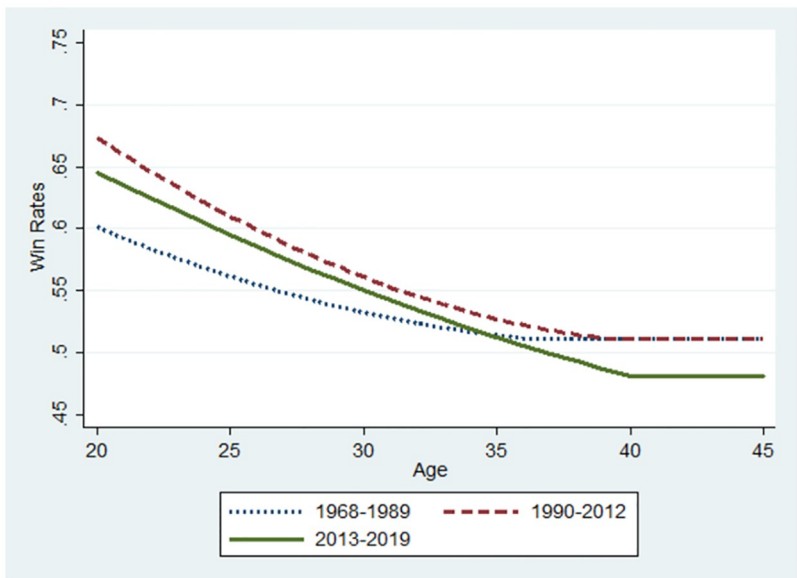

**Fig 12. Changes in performance using the sample of players with low ability.**

periods. Figs 11 and 12 jointly imply that the diffusion of AI drastically reduced performance as players aged, and this negative effect was larger for high innate-ability players.

In S1 Figs 2(1)-2(3) in the S1 File illustrate how effect of ages on winning rate varied according to debut ages. In S1 Fig 2 in the S1 File, we visualized how experiences covered lack of innate ability. Surprisingly, we observe positive relation between effect of ages and debut ages in the period 1968–1989, whereas negative relation between them in the period 1990–2012 and the period 2013–2019. In our interpretation, before emergence of new technology, learning from experience reduce the gap of innate ability between players. However, new technology reduced the learning effect.

The strategy automatically recommended by AI is far better than the strategy that many players think. However, the high ability players' skills and strategies are better than low ability ones. Hence, low innate ability players can enjoy the benefit of AI more than high ability players. Adopting the strategy recommended by AI compensates for the skill gap between the low and innate ability players. This implies that AI is a substitute for the innate ability of the Professional Shogi players, reducing the performance gap between the high and low ability players.

In international chess, players' productivity increased rapidly to peak age at 21 years and then declined gradually and constantly [20]. Furthermore, many players dropped out of play before the peak age [20]. Several Shogi players have been forced to drop out before becoming professional players when they belonged to the Shoreikai. Therefore, on average, players debut as professional at 21 years old, equivalent to the peak age of international Chess players. Consistent with the case of international Chess, as illustrated by the predictions from the Probit model in Figs 9–12, the Shogi players winning rates declines constantly after they turn 21 years old. Before entering the professional league, players are thought to improve their skills and performance in the semi-professional Shoreikai league. However, they do not play with professional players, except for some exhibition matches.

The finding of this study that Shogi players' performance declined consistently is in line with the findings of international chess studies that the age-productivity relationship is an inverted U-shape profile, with a peak at approximately 21 years for the case of chess [20]. The relation of FI drivers' age-productivity relationship is also an inverted U-shape. However, its peak is at approximately 31 years [31]. Interestingly, the mental productivity of the Shogi and Chess players declined by the age of 10, younger than that of FI drivers.

## Section 6: Discussion

Shogi and Chess share various similarities. However, Shogi is more complicated than Chess due to the following differences. First, the size of the board and the number of pieces are larger in Shogi than Chess. In chess, the board is 9x9 squares and 8x8 squares for Shogi and Chess, respectively. There 20 and 16 pieces for each player at the start of the game for Shogi and Chess, respectively. Second, there is difference in the way of captured pieces are used. In chess, by either player. In shogi, captured pieces can be returned to the board and used by the capturing player, whereas in Chess, pieces cannot be used once they are captured. Hence, there is more room for exerting player's creativity about strategies. Hence, creativity is more required for Shogi players to win the game than Chess. Different from Chess, Shogi game rarely ends in a draw. These makes Shogi games competitive because lower skilled players are less able to aim for a draw in Shogi than in Chess.

The innate ability is critical to win the game. Innate ability can be classified into twofold. First, players with higher innate ability are more able to surpass opponent through speedy calculation than other players. Second, they can have big-picture perspective based on intuition

to assess the situation better than others. However, AI reduces the gap of performance between those have innate ability and others. In compared to the period before emergence of AI, professional players need to spend enormous time on learning wide variation of strategies in grater details using AI. Any players hardly win if quantity of research is not sufficient to obtain the common knowledge of active players. It's just a starting point on the premise of extensive research.

The environment in which professionals are engaged in chess is becoming severer than ever before. The situation is reflected in interview with Akira Watanabe, the top Shogi players [32]. Watanabe said;

"I used to not do any research on Saturdays and Sundays because it was tiring. But now that's not enough time, and since around 2018, when I started to get serious about research in AI, I don't have the concept of 'taking weekends off'."

Answering to question 'Until how old do you plan to continue fighting through this kind of research?'

After thinking about it for a while, Watanabe said: "My image is that I will continue until I am about 45 years old. The degree of exhaustion in this way is completely different from that of the past. In the future, the life span of a professional player may become shorter."

Drastic change of work environment that Watanabe explained is congruent with the findings of our study.

For professional Shogi players, they are obliged to make their decision by themselves in the game and so using AI during game is against rule. However, this is the exceptional case. Generally, workers are encouraged to use AI for intellectual tasks to improve their performance and labour productivity because relation between AI and human ability is considered to be complement. Learning through experiences enables workers to reach right and desirable judgement. Hence, AI possibly increases performance of older workers who accumulated abundant human capital through experience.

## Section 7: Conclusion

The development and diffusion of AI result in drastic changes in the work environment in the professional Shogi world. In response to this, the professional Shogi players changed their work style to survive as professional players. This study quantitatively examined the impact of technological progress on the performance gaps between players using the games level-data covering the periods before and after the technological progress.

In 2018–2019, twin peaks appeared in the Shogi players' performance proving performance polarisation. In contrast, the performance is distributed in the form of unimodal distribution if we exclude the sample from 2018 to 2019. We found through regression analysis that (1) AI narrowed the gap of innate ability among same-age players leading to a reduction in the performance gap among them, (2) a consistent decline in the players' winning rate is observed as they got older, (3) the ageing decline of the probability of winning are observed in multiple periods and (4) the AI effects on the ageing decline of the probability of winning is only observed for high innate skill players. These findings suggest that the diffusion of AI caused players to retire from active play earlier in their career, whereas the gap in innate ability to survive as active players was reduced. That is, active players using AI can be considered a substitute for the innate ability of the individual player. These imply that the polarisation of players' performance depends on whether players can make the best use of AI rather than innate ability.

In 1996, Shuji Sato, a middle-class professional Shogi player, confessed, 'I only can pray that AI never wins against professional Shogi players because AI surpassing humans leads

demand for professional players to disappear' [7]. As opposed to his pessimistic view, AI actually made professional Shogi more popular and increased its demand. This is partly because AI numerically decides which player is better in the game. So viewers of the Shogi game can enjoy the game by predicting the winning probability even if they do not have any discernment. AI makes professional Shogi more attractive to everybody and a popular form of entertainment. The professional Shogi game has become like a professional sports game, such as a professional tennis game. There was a tag-match tournament where tag teams of professional Shogi players and AI participated. This agrees with the discussion that collaboration between humans and AI enhances creativity [5]. AI increases the demand for professional Shogi games. Meanwhile, the professional Shogi players came to retire earlier than ever before. It becomes more difficult to catch up with the speed of change of the newly established strategy if players get older. Naturally, players are more likely to be replaced by younger players. Hence, the mental productivity of the Shogi players is more likely to be physically productive like that of professional sports players.

What we observed is obtained under the setting of the labour market of professional Shogi where the use of AI during game is prohibited. However, using AI is desirable and even encouraged for many intellectual tasks in other works. As workers get older, they tend to have greater experience and judge more rightly in their works. Hence, AI possibly increases older workers' performance if AI is complement to judgement and experience. We should explore whether AI is an efficient tool in the hands of an experienced workers. It is unknown whether the effect of AI diffusion reduced the inequality caused by the innate ability in the settings of other intellectual tasks. Future research is needed to analyse how and the extent to which AI diffusion is a substitute for innate ability.

## Supporting information

**S1 File.**
(PDF)

## Acknowledgments

We would like to thank Editage (http://www.editage.com) for this manuscript's English language editing and review.

## Author Contributions

**Conceptualization:** Eiji YAMAMURA.

**Data curation:** Ryohei HAYASHI.

**Formal analysis:** Eiji YAMAMURA.

**Investigation:** Eiji YAMAMURA.

**Methodology:** Eiji YAMAMURA.

**Project administration:** Eiji YAMAMURA.

**Resources:** Eiji YAMAMURA.

**Software:** Eiji YAMAMURA.

**Supervision:** Eiji YAMAMURA.

**Validation:** Eiji YAMAMURA.

**Visualization:** Eiji YAMAMURA.

**Writing – original draft:** Eiji YAMAMURA.

**Writing – review & editing:** Eiji YAMAMURA.

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
