## [Decision Letter · Decision Letter 0]

7 Nov 2023

PONE-D-23-24861AI, Ageing and Brain-Work Productivity: Technological Change in Professional Japanese ChessPLOS ONE

Dear Dr. YAMAMURA,

Thank you for submitting your manuscript to PLOS ONE. After careful consideration, we feel that it has merit but does not fully meet PLOS ONE’s publication criteria as it currently stands. Therefore, we invite you to submit a revised version of the manuscript that addresses the points raised during the review process.

We look forward to receiving your revised manuscript.

Kind regards,

Claudia Noemi González Brambila, Ph.D.

Academic Editor

PLOS ONE

Reviewers' comments:

Reviewer's Responses to Questions

**Comments to the Author**

1. Is the manuscript technically sound, and do the data support the conclusions?

Reviewer #1: Yes

Reviewer #2: Yes

2. Has the statistical analysis been performed appropriately and rigorously? 

Reviewer #1: Yes

Reviewer #2: No

3. Have the authors made all data underlying the findings in their manuscript fully available?

Reviewer #1: Yes

Reviewer #2: No

4. Is the manuscript presented in an intelligible fashion and written in standard English?

Reviewer #1: Yes

Reviewer #2: Yes

5. Review Comments to the Author

Reviewer #1: This paper examines how technology and AI influence the aging and innate ability of Shogi players using and comparing records of three periods, from 1968 to 2019 according to technological development. Describes the rules of the Japanese chess game and the system to become professional players.

After conducting a thorough statistical analysis, a suitable model for representation has been developed. The conclusions drawn are satisfactory, although there may be further implications to consider. It is recommended to delve deeper into defining competitiveness, game complexity, creativity, and innate ability. What the authors suggest for Shogi and AI? What could be future scenarios?

Please review the following lines: 110, 470

Reviewer #2: There are some issues with the statistical analysis. But they are easily fixable.

The authors claim that the data is available. But the link points to a page in Japanese, so it is not easy figure out how to download the data. Comments are in the attached file.

6. PLOS authors have the option to publish the peer review history of their article (what does this mean?). If published, this will include your full peer review and any attached files.

Reviewer #1: No

Reviewer #2: No

---

## [Author Response · Author response to Decision Letter 0]

3 Jan 2024

Reviewer #1: This paper examines how technology and AI influence the aging and innate ability of Shogi players using and comparing records of three periods, from 1968 to 2019 according to technological development. Describes the rules of the Japanese chess game and the system to become professional players.

After conducting a thorough statistical analysis, a suitable model for representation has been developed. The conclusions drawn are satisfactory, although there may be further implications to consider. It is recommended to delve deeper into defining competitiveness, game complexity, creativity, and innate ability. What the authors suggest for Shogi and AI? What could be future scenarios?

Reply： We newly added the “6. Discussion” section to provide implications related to the points suggested by the referees.

Please review the following lines: 110, 470

Reply： We fixed the lines as follows,

 112: the number of section “2.2” was revised to “2.1”.

 470: Expression is fixed.

Reviewer #2: There are some issues with the statistical analysis. But they are easily fixable.

Your comments are very valuable in substantially improving the quality of the paper. I hope that you find it satisfactory—please find our replies to your comments below. For the convenience of review, the revised parts are colored in red in the paper, and the replies to the referee are in blue. I have submitted the data and program files together with the revised manuscript and Figures files.

Clarification.

• Through all the paper, the phrase “the novel setting” is used. The authors must clarify

what do they mean by that, or find a different wording.

Reply 1 to Clarification: We deleted “novel” and added a phrase to explain the setting;

Page 2. Line 21: “the novel setting” (former version)

Page 2. Line 20-21:”the setting where various external factors are controlled in deterministic and finite games” (Revised version)

Page 5. Line88: “the novel setting” (former version)

Page 5. Line 80: “the setting in deterministic and finite games” (Revised version)

Page 18. Line 303: “Under the novel setting of” (Former version)

Page 18. Line 284: “Under the setting of” (Revised version: only deleting “novel”)

Page 42, Line 697: “the novel setting” (former version)

Page 42, Line 654-655: “the labor market of professional Shogi, where the use of AI during games is prohibited.” (revised version).

• Page 2 line 26: “diffusion of artificial intelligence (AI) reduces innate ability”. I think

that the authors really meant to say “diffusion of artificial intelligence (AI) reduces

the impact of innate ability in player’s performance”.

Reply 2 to Clarification: 

Page 2 line 26: “Following the comment, “diffusion of artificial intelligence (AI) reduces innate ability” is replaced by “diffusion of AI reduces the impact of innate ability on player performance”.

• Authors should be more explicit on which of their finding are specific to board games

as sports (Chess,Shogi,Go,etc.), to Shogi in particular, or to intellectual labor in

general. For example, finding (1) in the Abstract could be extrapolated to the labor

market, but I am not so sure about findings (2)-(4). After all, these are deterministic

and finite games, the outcomes of which lack any ethical implication.

Reply 3 to Clarification: findings (2)-(4) were changed as below.

Page 2. Line 28-32: ((2) in all the periods, players’ winning rates declined consistently from 20 years and as they got older; (3) AI accelerated the aging decline in the probability of winning, which increased the performance gap among different-aged players; (4) effects of AI on the aging decline and the probability of winning were observed for high innate-skill players but not for low innate-skill players. 

• Philosophizing a bit, I would say that AI might be more beneficial for older workers

(not less) in many areas. Taking Chess as an example, we can argue that as players

get older they tend to assess positions better due to experience, but they get slower

calculating variations. That is, as “judgment” and “experience” increases, the “raw

brain power” decreases. However, in board sports the use of AI during the game is

cheating, but for many intellectual tasks it is desirable and even encouraged. While

current AI cannot substitute human “judgment” and “experience” in many areas, it

can greatly help with the “raw brain power”. Hence I argue that, in many examples,

AI may be a more efficient tool in the hands of an experienced worker.

Reply 4 to Clarification: We argue the points in the newly added “6 Discussion” and in the last paragraph of 7 Conclusion.

pp.37-40. Discussion section.

Page 42. Lines 654-659. “where the use of AI during games is prohibited. However, using AI is desirable and encouraged for many intellectual tasks in other works. As workers age, they tend to gain greater experience and judge their work more correctly. Hence, AI may improve older workers’ performance if AI complements judgment and experience. Therefore, exploring whether AI is an efficient tool for experienced workers is necessary.”

• Page 3 line 51: “However, the level of AI is almost equivalent”. I guess it was meant

to say “However, the level of AI was almost equivalent” (at the time of Watanabe’s

prediction).

Reply 5 to Clarification: 

Page 4. Line 51. Following this comment, we revised the sentence.

• Page 4 line 54: “games against AI”. Shouldn’t it say “games against other human

players?”.

Reply 6 for Clarification: This is a careless mistake. Thank you for your comment. Following this comment, we revised the sentence.

Page 4. Line 57. “games against other human players” (revised version)

• Page 4 lines 68-69: “The recent development of AI technology allows it to replicate

the human brain and, therefore, would replace brain work.” I strongly disagree.

Current AI gives us only the illusion of intelligence. It excels at pattern recognition,

but it cannot reason, and does not understand concepts. For example, given

thousands images to train, AI can distinguish a horse form a unicorn. In contrast, a

child can figure that “a unicorn is simply a horse with a horn” with less than 10

images. Current AI cannot figure out that simple fact. In Chess/Shogi AI finds good

strategies, but so far only humans can rationalize why those strategies work, and apply

them accordingly. Therefore, in the near future, AI will replace only “some

intellectual work”, because it cannot replicate our reasoning yet.

Reply 7 to Clarification: We argue the points in the newly added “6 Discussion” and the last paragraph of 7 Conclusion. See Reply 4 for further clarification.

• Page 6 lines 96-97: “AI is a substitute for the innate ability of an individual player”.

I would soften a bit such a strong assertion. For example, “AI training reduces the

importance of the innate ability of an individual player”.

Reply 8 to Clarification. Following the comment, we changed the sentence.

Page 6, lines 98-99: “AI training reduces the importance of an individual player’s innate ability” 

• Page 10 lines 177-178: “No female Shogi players can win through in Shoreikai and

be promoted to become professional players”. It is not clear from the exposition if

women are not able to win because of their lack of skill, or if they are simply not

allowed to participate in Shoreikai.

Reply 9 to Clarification. We added the sentence below:

Page 11 line 176-177: “Female players are not able to win because of their lack of skills.”

• Page 12 lines 203-206: I do not follow the reasoning why “the degree of dan is

unlikely to reflect the players’ strength”. As I understand, this degree

increases/decreases with good/bad performance. If performance is a proxy for

strength, then why the degree of dan is not also a proxy for strength? All this is related

with the phrase “Elo rating can be a player’s ability that differs from performance

(Bertoni et al. 2015)” found in lines 359-360, page 21. It is important that the authors

clarify how performance and ability/strength differ.

Reply 10 to Clarification: “Elo rating can be a player’s ability that differs from performance.

(Bertoni et al. 2015)” has been deleted. Concerning degree of dan, we have added the following explanation:

Page 13, lines 205-208. “Meanwhile, the degree of dan never decreases once the dan is certified to the player. That is, the degree of dan is fixed, even if the player’s winning ratio is drastically reduced. Therefore, the degree of dan is unlikely to reflect players’ current strengths.”

• Page 15 line 257: “Its form is similar to the normal distribution for the periods …”.

The phrase in the text is not wrong, but it suggests normality. However, normality is

not tested, and not directly needed, for the rest of the paper. I suggest avoiding

mentioning normality. For example “We observe a unimodal distribution for the

periods …”. The same happens in page 34 lines 570-571.

Reply 11 to Clarification: Following the comment, we have replaced the former sentence with the sentence suggested by the referee. 

Page 15, Line 257-258. Page 37, Line 626.

• Page 16 line 279: “Sample covers ages over 45 and ...”. I suggest changing it to

“Sample covers players of alages, and …”.

Reply 12 for Clarification: Following your comment, we have changed the phrase. Page 16, Line 278-279. 

Methodology:

• The Probit model should be better presented and interpreted, for instance …

Reply 1 to Method: We have misunderstood what is reported in the tables due to the lack of careful explanation. In all tables, we reported “marginal effects” but not “coefficient.”

We newly added the following explanation to avoid the reader’s misunderstanding. 

Page 24. Lines 409-413. 

“The ordinary least squares (OLS), coefficients, and marginal effects are the same. However, the probit model used in this study differs. Tables 2‒4 show the probit model results. The values without parentheses in each variable are marginal effects but not coefficients. We used Stata 14 for these estimations, which is widely used in economics. In the program, “dprobit” was used to obtain the marginal effect.” 

The referee suggested the following command of Stata to obtain the marginal effects.

probit Y X

margins, dydx(*) atmeans

We obtained the same results using another command because the estimation results can be obtained more quickly than as suggested by the referee.

dprobit Y X 

In addition to the Manuscript, Figures, and this explanation, we have submitted the Data and Stata Command. 

• … in Page 21 lines 355-357: Instead of “WINigt = α0+…+X’itB +ui”, it must say

something like “Prob[WINigt = 1] = F(α0+…+X’itB), where F is the cumulative

distribution function of the standard normal distribution”.

Reply 2 to Method: 

Page 21. Lines 356-360: We revised the mathematical expression to meet your requirements.

• Page 22 lines 374-377: “Therefore, we use the winning … variable is 0 or 1.”. This

explanation should come before the functional form of the model presented in lines

355-357.

Reply 3 to Method: Page 21. Lines 351-354: Following this comment, we relocated the sentences.

• My main problem with the use of Probit is that it is not clear the distinction between

the coefficients of the model, and the marginal effects. In an OLS the two things are

the same. But Probit is not linear (even without the quadratic terms in age) because

of the cumulative function F. Specifically …

Reply 4 to Method: See “Reply 1 to Method”.

• … in Page 25 lines 419-420 it says that “Numbers without parentheses are marginal

effects”. I believe that this is not the case. I guess that in Table 2 you are showing

the coefficients (alphas) of the equation in lines 355-357. What is important here is

to show which alphas are statistically significant, and their signs. But to interpret them

as marginal effects is not correct. The marginal effect is the derivative of the

Prob[WINigt = 1] with respect to one variable, fixing the rest at some specific values.

Reply 5 to Method: See “Reply 1 to Method”.

• Actually, it would be very desirable to report another table with the actual marginal

effects, keeping the rest of variables at their mean values. These marginal effects then

can be interpreted as the probability change for a given change in the respective

variable.

Reply 6 to Method: See “Reply 1 to Method”.

• Page 25 line 432: “winning rate is negative and linear othing s linear in the robit

model (again, because of function F). At most, you can say that “winning rate is

decreasing regardless of the other variables”. This not e if the coefficient

of the quadratic age were statistically different from zero. Please, rephrase all your

interpretations correctly.

Reply 7 to Method: 

Page 27. Lines 436. We replaced “winning rate is negative and linear” with “winning rate was

decreasing, regardless of the other variables”. Also See “Reply 1 to Method.”

• Page 27 line 457: change the name of the section from “Simulation” to something

like “Predictions from the Probit model”. This section is about the predicted

probability of winning from the model estimated in the previous section. Therefore,

avoid any mention to “simulation” through the whole paper.

Reply 8 to Method: Following the comments, we changed the name of the section from “Simulation” to “Predictions from the Probit model.” We did this throughout the manuscript.

Page 23. Line 400

Page 29. Line 469, 471, 481

Page 31. Line 513

Page 34. Line 562

• Page 30 line 517: “Panel A. Low innate ability”. I think it should be ‘High’.

Reply 9 to the Methods section: We thank you for your suggestion. “Panel A. Low innate ability” was changed to ‘Panel B. High innate ability’. Accordingly, Figure 12 was changed to Figure 11 in the Figures file and in the main body of the text.

• Page 31 line 518: “Panel B. High innate ability”. I think it should be ‘Low’.

Reply 10 to the Methods section: Thank you for your suggestion. “Panel B. High innate ability” is changed to ‘Panel B. Low innate ability’. Figure 11 was changed to Figure 12 in the Figures file and the main body of the text.

Page 28. Line 463

Page 32. Line 535.

• The explanations at lines 470 and 473 in page 28 are illustrative of the intuition behind

the quadratic terms, but are not entirely accurate (because the function F is being

ignored).

Reply 11 to Method: We calculated the values and illustrated Figures based on marginal effects using the command “dprobit.” 

• Assuming that you are using Stata for your analysis, the following “toy” example

illustrates what I am asking for:

probit WIN c.AGE##c.AGE c.DEB_AGE##c.DEB_AGE c.ELO c. ELO_OP (and other vars.)

margins, dydx(*) atmeans

margins, dydx(DEB_AGE) at(DEB_AGE=(16(1)26))

marginsplot

margins, dydx(AGE) at(AGE=(20(1)60))

marginsplot

(please remember to put a i. in front of dummy/categorical variables, and c. in front

of any continuous variable)

Reply 12 to Method: Following the comment, we have provided Figures B1, B2, and B3 in a supplementary file, which we illustrated using the suggested command. In the Figures, the slopes are too gentle to interpret because the confidence intervals (CI) are large. We have checked the statistical significance in the tables, although statistical significance was not observed by CIs in Figure. In any case, what is illustrated in Figures B1, B2, and B3 is already shown in Figures (former version), where we put the three lines jointly, although the CIs are not shown. We present figures for readers to visualize and compare the effects visually rather than to check statistical significance. 

Figures B1, B2, and B3 are not informative. Therefore, we only provided Figures B1, B2, and B3 for the referee to check but did not add them to the Appendix to avoid duplication.

• Also, I would like to recommend you to explore the interaction between AGE and

DEB_AGE (to see is the decline is different for different levels of innate ability).

Something like this:

probit WIN c.AGE##c.AGE c.DEB_AGE##c.DEB_AGE c.AGE##c.DEB_AGE (etc.)

margins, dydx(AGE) at(DEB_AGE=(16(1)26))

marginsplot

Reply 13 to Method: Following the com

---

## [Decision Letter · Decision Letter 1]

18 Jan 2024

PONE-D-23-24861R1AI, Ageing and Brain-Work Productivity: Technological Change in Professional Japanese ChessPLOS ONE

Dear Dr. YAMAMURA,

Thank you for submitting your manuscript to PLOS ONE. After careful consideration, we feel that it has merit but does not fully meet PLOS ONE’s publication criteria as it currently stands. Therefore, we invite you to submit a revised version of the manuscript that addresses the points raised during the review process.

We look forward to receiving your revised manuscript.

Kind regards,

Claudia Noemi González Brambila, Ph.D.

Academic Editor

PLOS ONE

Journal Requirements:

Reviewers' comments:

Reviewer's Responses to Questions

**Comments to the Author**

1. If the authors have adequately addressed your comments raised in a previous round of review and you feel that this manuscript is now acceptable for publication, you may indicate that here to bypass the “Comments to the Author” section, enter your conflict of interest statement in the “Confidential to Editor” section, and submit your "Accept" recommendation.

Reviewer #2: (No Response)

2. Is the manuscript technically sound, and do the data support the conclusions?

Reviewer #2: Partly

3. Has the statistical analysis been performed appropriately and rigorously? 

Reviewer #2: No

4. Have the authors made all data underlying the findings in their manuscript fully available?

Reviewer #2: Yes

5. Is the manuscript presented in an intelligible fashion and written in standard English?

Reviewer #2: Yes

6. Review Comments to the Author

Reviewer #2: (No Response)

7. PLOS authors have the option to publish the peer review history of their article (what does this mean?). If published, this will include your full peer review and any attached files.

Reviewer #2: No

---

## [Author Response · Author response to Decision Letter 1]

1 Feb 2024

PONE-D-23-24861R1

AI, Ageing and Brain-Work Productivity: Technological Change in Professional Japanese Chess

PLOS ONE

Dear Prof. Claudia Noemi González Brambila:

Thank you for your and the referees’ positive evaluations of this paper. We have revised and substantially improved the manuscript in accordance with the referees’ insightful comments and suggestions. 

For your convenience, our replies to the referee are shown in blue.

The comments we received from you and Referee 2 were very helpful. We hope the reviewers find our replies to these comments satisfactory. 

Best regards,

Eiji YAMAMURA

Reviewer #2: 

One important methodological aspect remains to be fixed, and it has to do with the marginal effects in the Probit model.

Thank you for the detailed instructions on how to calculate the marginal effects of the probit model. I learned a lot and improved the quality of this paper through your comments. I hope you find the revision satisfactory. Below are our replies to your comments. For your convenience, the revised parts are colored red in the paper, and the replies to the referee are shown in blue. I have submitted the data and program files together with the revised manuscript and the figures files.

For your convenience, I have also numbered each comment.

1. Eliminate all squared variables from Table 1 (like AGE2/100), because they are not

independent variables.

Reply to 1: In compliance with this comment, we have deleted all the squared variables from Table 1.

2. Correct Tables 2-4. In each case, I would show two tables instead of one: the actual coefficients (the output of the ‘probit’ command), and then the marginal effects

(the output of the corresponding ‘margins, dydx(*)’ command). The first table

contains the quadratic terms, but the second table does not.

Reply to 2: The previous version of Table 4 consisted of Panels A and B. In the revised version, the contents of Table 4 have been divided into Tables 4 (a subsample of high ability) and 5 (a subsample of low ability). To meet the reviewer’s requirements, the new Tables 2-5 consist of Panels A (coefficients) and B (marginal effects).

3. When reporting the marginal effects, it is required to also report at which values of

the variables they have been computed (by default, at their means).

Reply to 3: The marginal effects of the variables on their means are reported, as suggested by the reviewer. In addition, we reported a marginal effect of “Age” at 35 years old. We added this sentence in the notes section below Tables 2-5, “In Panel B, the marginal effects are evaluated at mean values for the independent variables, except for the bottom line, where the marginal effect of Age is evaluated at 35 years old.”

.

4. Check that Figures 7-12 are generated correctly, taking into account all the

nonlinearities of the Porbit model. As mentioned before, Stata ‘margins’ and

‘marginsplot’ commands can do that, but I do not know a simple way to overlap the

predictions for the three time periods into a single graph. At least check that your

figures are the same as the ones Stata produces.

Reply to 4: Based on the newly estimated marginal effects on winning at various variable values, we calculated the “level of wins” to illustrate the new Figures 7-12. The program and dataset for the figures are attached. 

5. In page 24 lines 409-410 I would remove “The ordinary least squares (OLS),

coefficients, and marginal effects are the same. However, the probit model used in

this study differs.”. As explained before, this is not true for all OLS regressions.

Reply to 5: Based on your comment, we have deleted this sentence.

6. In page 21 line 360: change “standard norm distribution” by “standard normal

distribution”.

Reply to 6: Thank you for pointing out this error. We fixed it.

7. Revise all the explanations and interpretations resulting from the previous changes

Reply to 7: We have revised the main body of the text to explain and interpret the new Tables 2-5, Figures 7-12, and Appendix Figures A2-A7. The main revisions are as follows:

(1) We explained how to calculate the marginal effect, which is heavily dependent upon your instructions (Lines 401-417; Lines 428-437).

(2) In 5.1., we interpreted the results of the Tables (Lines 493-503).

(3) A new interpretation of Figures 7-12 can be found in Section 5.2. (Lines 526-595).

---

## [Decision Letter · Decision Letter 2]

19 Feb 2024

AI, Ageing and Brain-Work Productivity: Technological Change in Professional Japanese Chess

PONE-D-23-24861R2

Dear Dr. YAMAMURA,

We’re pleased to inform you that your manuscript has been judged scientifically suitable for publication and will be formally accepted for publication once it meets all outstanding technical requirements. (I'm sorry it could not be sooner, I did my best). 

Kind regards,

Claudia Noemi González Brambila, Ph.D.

Academic Editor

PLOS ONE

Additional Editor Comments (optional):

Reviewers' comments:

Reviewer's Responses to Questions

**Comments to the Author**

1. If the authors have adequately addressed your comments raised in a previous round of review and you feel that this manuscript is now acceptable for publication, you may indicate that here to bypass the “Comments to the Author” section, enter your conflict of interest statement in the “Confidential to Editor” section, and submit your "Accept" recommendation.

Reviewer #2: (No Response)

2. Is the manuscript technically sound, and do the data support the conclusions?

Reviewer #2: Yes

3. Has the statistical analysis been performed appropriately and rigorously? 

Reviewer #2: Yes

4. Have the authors made all data underlying the findings in their manuscript fully available?

Reviewer #2: No

5. Is the manuscript presented in an intelligible fashion and written in standard English?

Reviewer #2: Yes

6. Review Comments to the Author

Reviewer #2: Only minor changes are needed now. Please see the attached comments.

7. PLOS authors have the option to publish the peer review history of their article (what does this mean?). If published, this will include your full peer review and any attached files.

Reviewer #2: No
